# The Effects of Mixed Foliar Nutrients of Calcium and Magnesium on the Major Bypass Respiratory Pathways in the Pulp of 'Feizixiao' Litchi

Shaopu Shi [1,2], Jingjia Du [2], Junjie Peng [1,2], Kaibing Zhou [1,2,*] and Wuqiang Ma [1,2,*]

1   Sanya Institute of Breeding and Multiplication, Hainan University, Sanya 572025, China; ssp@hainanu.edu.cn (S.S.); pjj573562838@163.com (J.P.)
2   College of Tropical Agriculture & Forest, Hainan University, Danzhou 571737, China; djjhndx123@163.com
*   Correspondence: zkb@hainanu.edu.cn (K.Z.); wuqiangma@hainanu.edu.cn (W.M.)

**Abstract:** During the period of 'Feizixiao' litchi fruit pericarp's full coloring, there is a phenomenon of "sugar withdrawal" in the pulp, and the mixed foliar nutrients of calcium and magnesium (Ca+Mg) can effectively overcome this phenomenon. One of the reasons for this may be that it is related to the influence of the mixed nutrients of Ca+Mg on the bypass respiratory pathways of the pulp. The major fruit quality indicators, the rates of cytochrome and cyanide-resistant respiratory pathways (CP and AP) in the pulp and the activities of their key enzymes, were observed continuously in 2021 and 2022, and the deferentially expressed genes (DEGs) related to the two bypass respiratory pathways in the pulp were screened by RNA-seq analysis, with a qPCR of the random genes performed to verify the results. Ca+Mg treatment kept the content of the total soluble sugar in the pulp stable and higher than that the control in the ripening stage; Ca+Mg treatment increased the activities of electron-transferring enzymes in the electron transport chain, such as NADH dehydrogenase (ND), succinate dehydrogenase (SDH), cytochrome bc1 complex, and cytochrome c (Cyt c) through up-regulating their gene expression. In terms of the rate-limiting enzymes in the pulp, Ca+Mg treatment increased the activity of cytochrome oxidase (*COX*) in the CP pathway by up-regulating the expression of *COX* genes, then increased the CP respiratory rate and inhibited the CP respiratory rate decrease; meanwhile, it also inhibited the activity of *AOX* (alternate oxidase) in the pulp in the AP pathway by down-regulating the expression of *AOX* genes, then inhibited the increase in the AP respiration rate. The qPCR validation of randomly selected DEGs showed a significant unitary linear correlation between their expression levels and the results of the RNA-seq analysis. Therefore, one of the physiological mechanisms on the mixed foliar nutrients of Ca and Mg overcoming the phenomenon of "sugar withdrawal" in the 'Feizixiao' litchi pulp could be to promote CP and to inhibit AP, and then to delay the ripening and senescence of the pulp.

**Keywords:** sugar withdrawal phenomenon; bypass breathing pathway

## 1. Introduction

*Litchi chinensis* cv. Feizixiao is the main cultivar in the litchi-producing regions of China. Although it is famous for its fruit flavor, the pulp becomes acidic when the pericarp is fully red and the pericarp colors poorly when the pulp tastes the sweetest. This phenomenon is known as "sugar withdrawal" of the pulp or "stagnant green" of the pericarp. As a result, the market competitiveness and economic benefits of 'Feizixiao' litchi decrease [1,2].

In regard to the development of the quality of 'Feizixiao' litchi fruit, numerous studies have been conducted on the mechanism of the "stagnant green" of the pericarp, and some progress has been made. During the growth and development of 'Feizixiao' Litchi fruits, the issues of the pericarp's "stagnant green" can be overcome by fruit bagging, which due to the fact that the activity of flavonoid glycosyltransferase (UFGT) is positively correlated with the accumulation of anthocyanin in the pericarp; meanwhile, fruit bagging treatment can

inhibit the activity of UFGT in the pericarp, thus reducing the accumulation of anthocyanin in the pericarp. Therefore, the UFGT activity and content of anthocyanin in the pericarp both increase rapidly after removing the bag [3]. The foliar nutrient of Mg can overcome the issue of "stagnant green" of 'Feizixiao' litchi pericarp after fruit setting. The foliar nutrient of Mg can raise the content of ABA in pericarp and improve the activity of UFGT, thus increasing the accumulation of anthocyanin [4].

In recent years, reports of research on the physiological and biochemical mechanism of pulp's "sugar withdrawal" and the measures for overcoming it have increased more and more. However, further in-depth research on it is still required. The foliar nutrient of a mixture of Ca and Mg inhibits the aerobic respiration of the EMP-TCA pathway through decreasing the activities of GPI, SDH, and NADP-ME in the pulp of 'Feizixiao' Litchi, then reduces the loss of sugar in the pulp (Du Jingjia et al., 2023) [5]. Simultaneously, the foliar nutrient of the mixture of Ca and Mg down-regulates the expression of the structure genes of sucrose synthase (SS) and sucrose phosphate synthase (SPS) in pulp and the sucrose in pulp is inhibited by synthesizing, so the accumulation of hexose is promoted. On the other hand, the expression of fruit hexokinase (HK) structural genes is down-regulated by the foliar nutrient of the mixture of Ca and Mg and it inhibits the hexose flow to the EMP and the PPP, then increases the accumulation of hexose, so the issue of the pulp's "sugar withdrawal", as well as the problem of "fruit loss", are both overcome (Peng J et al., 2022) [6].

However, the plant's respiratory system is a set of mighty and complex metabolic networks, which encompass the electron transfer chain except for EMP, EMP-TCA, and PPP [7]. The electron transport chain is created through the transfer of electrons between redox substrates, resulting from the loss or gain of electrons from carriers such as ubiquinone due to oxidative phosphorylation. This is an in vivo reaction that oxidizes substances to release energy for synthesizing ATP from ADP and inorganic phosphate via the respiratory chain [8]. Mitochondria in higher plants exhibit two primary chains for electron transfer. The first is the transportation of electrons from ubiquinone to oxygen via the cytochrome oxidase (*COX*) pathway, known as CP, which generates adenine nucleoside triphosphate (ATP). The second is through the alternative oxidase (*AOX*) pathway, designated as AP, which does not generate ATP [9].

The CP pathway is a crucial stage in the oxidation reaction at the end of the electron transport chain of mitochondrial respiration. It transports $H^+$ to generate ATP [10]. The electron transport chain encompasses four membrane-bound oxidoreductase complexes: NADH dehydrogenase (complex I), succinate dehydrogenase (SDH), cytochrome reductase (P450), and cytochrome oxidase (*COX*). These complexes transfuse the electrons from NADH or succinate to $O_2$. Where cytochrome oxidase plays a crucial role in the oxidation process at the end of the mitochondrial respiratory chain, it catalyzes the transfer of electrons from reduced cytochrome C to $O_2$, the ultimate acceptor of electrons [11].

The AP pathway is a respiratory pathway that is insensitive to cyanide. This means that, when anions like cyanide are present in the plant body, the organism can respire using the cyanide-resistant pathway, which separates from the ubiquinone of the CP pathway. The electrons are then transferred directly to *AOX* to reduce molecular oxygen to water, a crucial process for both material and energy metabolism in the cell [12,13]. Alternate oxidase (*AOX*), also referred to as cyanide-resistant oxidase, is the terminal oxidase of the cyanide-resistant respiratory pathway. It transfers electrons from panthenol to oxygen, leading to water production and reduction in the transmembrane proton potential. *AOX* is resistant to cyanide but vulnerable to inhibition by salicyl oxime acid [14]. When the cytochrome pathway is restricted, alternative oxidase (*AOX*) is induced to receive electrons directly from ubiquinone and reduce oxygen to water, thus maintaining respiration [15]. AP pathway respiration is usually induced in plants when they encounter adversity [16–18].

The electron transport pathway also plays a role in fruit ripening and senescence. During the ripening process of fruits, they are divided into "climacteric fruits" and "non climacteric fruits" based on the trend of respiratory rate changes. Climacteric fruits are

characterized by a rapid increase in respiratory rate, usually accompanied by an explosion of ethylene; non-climacteric fruits do not exhibit significant respiratory peaks during their ripening process [19]. Climacteric fruits sometimes exhibit increased respiration during ripening, potentially linked to the AP pathway of the mitochondrial electron transport chain. Ripening climacteric fruits have higher respiration rates towards the end of their ripening period, but lower total respiration rates and a pronounced increase in the AP pathway [20]; non-climacteric fruits have increased *AOX* enzyme activity during ripening [21,22]. This physiological event reduces ATP synthesis, increases heat production, and promotes fruit ripening. Both the CP pathway and the AP pathway are subject to feedback regulation. Researchers discovered that the respiration rate of the CP pathway and *COX* enzyme activity decreased jointly in the early development of the ATCOX10 mutant Arabidopsis. Furthermore, the total respiration rate remained constant, while the respiration rate of the AP pathway increased significantly [23]. Additionally, during the ripening of non-respiratory leptotropic tomato fruits, *AOX* demonstrated increased expression. However, the *COX* II subunit of the *COX* enzyme decreases during ripening [22], and there is also a significant decrease in *COX* enzyme activity and content in bell peppers during late ripening [24]. Studies on the senescence of non-climacteric fruits have shown that the ATP content in the fruit is a critical factor that affects fruit senescence. A higher ATP content is advantageous in delaying fruit senescence and quality deterioration. Moreover, a higher *COX* enzyme activity and CP pathway respiration rate are favorable for ATP synthesis, thereby delaying fruit senescence [25–28].

The bypass respiratory pathway in the pulp of 'Feizixiao' litchi should also affect ripening and senescence, which was determined to be related to the phenomenon of the pulp's "sugar withdrawal". In the early stage of the study, the research group investigated the physiological mechanism of the aerobic respiration of the pulp by observing the effects of single and mixed foliar nutrient treatments of calcium and magnesium on the aerobic respiration [29]. In this paper, the research group further investigated the effects of the mixed foliar nutrient treatment of calcium and magnesium on the main bypass respiratory pathway in the pulp and explored its preliminary molecular mechanism, which will further enrich and perfect the theory of 'Feizixiao' litchi fruit growth and cultivation, laying a theoretical foundation for the genetic improvement in and cultural optimization of some litchi cultivars with poor pericarp coloration.

## 2. Materials and Methods

### 2.1. Experimental Materials

Ten 'Feizixiao' litchi adult trees with moderate growth, a uniform tree shape, no mechanical damage, no pests or diseases, and a high yield were selected on 18 April 2021 and 16 April 2022 in the litchi orchard of Jinpai Farm in Lingao County of Hainan Province. The orchard is located in the tropical monsoon zone with a temperature of 23–24 °C, average annual sunshine of 2175 h, an average annual precipitation of 1100–1800 mm, and synchronization of rain and heat. The soil belongs to brick red soil. The main phenological periods are as follows: February–March is the flowering period, physiological fruit drop starts in early April, the fruit expansion period is entered in late April, and the fruit ripening period is entered in mid-May.

### 2.2. Experimental Design Method

The 10 experimental trees were divided into 2 groups, and 1 was used for the treatment, which was dealt with the foliar nutrient of a 0.3% $CaCl_2$ and 0.3% $MgCl_2$ mixed aqueous solution (Ca+Mg), and the other was used as the control, which was dealt with clean water (CK). Single-plant plots and 5 replications were designed [30–32].

### 2.3. Field Treatment and Sample Collection Methods

On the 35th day after flowering, the seed was fully covered with the peridium and the fruit bottom became red, meaning that the physiological fall of fruits had ended and the

period of rapid fruit expansion had begun. Then, the field treatment was started, and from then on, the treatments were conducted 3 times each week. The treatment was carried out during the period from 9:00 a.m. to 10:00 a.m. The first sampling started before the first field treatment, and the sample fruits were collected from around the middle of the outer crown of each experimental tree's canopy. At the first sample time, 5 medium-sized fruits around the middle of the outer crown of each experimental tree's canopy were selected to be used as the reference fruits for dynamic sample collection, and these reference fruits were marked with a hang tag. The fruit sampling time lasted until the pericarp became fully red, namely, the sampling time lasted 70 d after flowering. In total, 30 sample fruits were collected per tree according to the size and color of the reference fruits from around the middle outer crown of the canopy each time, and the sample fruits were frozen with liquid nitrogen quickly in the field, then brought back to the laboratory and stored in the $-80\,°C$ ultra-low-temperature freezer refrigerated on standby.

*2.4. The Assay Methods of the Indicators*

### 2.4.1. Determination of the Pulp's Conventional Quality

The soluble sugar content of the fruit pulp was determined by the method of Wang H.C et al.) [33] with a slight modification: we weighed 0.5 g of pulp in a mortar, heated it in a microwave oven for 30 s, added 5 mL of 90% ethanol, ground the sample thoroughly, centrifuged it at $10,000\times g$ for 15 min, aspirated the supernatant, and added 5 mL of 90% ethanol for another extraction. Then, we combined the two supernatants, which were evaporated in a 90 °C water bath, Then, the remaining sample was fixed with 10 mL of deionized water, and a small amount was aspirated with a syringe and filtered through a 0.45 mm membrane for testing. The sugar contents were determined on a Waters 2695 high-performance liquid chromatograph with an evaporative light scattering detector and a Boston Green Amino Column ($4.6 \times 250$ mm, 5 mm). The titratable acid content was measured by sugar-acid meter, and citric acid was used as the calculation standard for the test. A portable colorimeter was used to measure the peel coloring a and b values, and then to find out the peel chromaticity angle (h).

### 2.4.2. Measurement of Pulp Respiration Rate

The determination of respiration rates through the CP and AP pathways in the fruit pulp was performed using a portable $O_2/CO_2$ headspace analyzer (Dansensor Check-Point3) [34]; Unit: $mL/CO_2kg^{-1}h^{-1}$ [35]; 1 mmol/L of potassium thiocyanate was used as an inhibitor of the CP pathway and 3 mmol/L salicyloxime acid was used as an inhibitor of the AP pathway. The measurement time was around 10:00 am and the ambient temperature was maintained at around 28 °C. The total respiratory rate of the sample was measured first each time. After vacuum infiltration of the inhibitor, the remaining respiratory rates of each pathway were measured. The observation value of the respiratory rate of a single pathway was obtained by subtracting the remaining respiratory rates of each pathway from the total respiratory rate [36,37].

*2.5. Transcriptome Analysis and Fluorescence Quantitative PCR Validation*

### 2.5.1. Differential Gene Screening Analysis

According to the existing transcriptome [6], differential genes were screened based on FDR < 0.05 and $|\log2 (FC)| \geq 1$. The selected genes were all related to electron transfer chains and cyanide-resistant respiration.

### 2.5.2. Primer Design and qRT-PCR Validation

Ten differentially expressed genes were selected, and real-time PCR primers, synthesized by Shanghai Bioengineering Co., Ltd (Shanghai, China)., were designed with Prime3 (https://primer3.ut.ee, accessed on 30 June 2023). Fruit pulp RNA was extracted using the Plant RNA Extraction Agent Set (Beijing Kulebo Technology Co., Ltd., Beijing, China) The extracted pulp RNA was reverse-transcribed into cDNA using a cDNA synthesis Kit

from Vazyme Biotechnology Co., Ltd. (Nanjing, China) and a T100FM Thermal Cycler PCR instrument from BIO-RAD (Hercules, CA, USA). The experiment was performed according to the instructions provided in the kit. Real-time PCR verification was performed with Taq Pro Universal SYBR qPCR Master Mix (Vazyme Code: Q712-02) from Vazyme Biotechnology Co., Ltd. (Nanjing, China) and a qTOWER3 instrument (Jena, Germany). Litchi actin was used as the internal reference gene. The primers utilized in the study are presented in detail in Table 1.

**Table 1.** Primer table.

| Primer Names | Left Primer Sequences (5′ to 3′) | Right Primer Sequences (5′ to 3′) |
|---|---|---|
| Cluster-6206.61154 | CGGACACTGCTTTACCCACT | TCACCCTTAGATGCCAAGCA |
| Cluster-6206.84300 | TTGTTCATGCGGCAACTTGG | GCCTCAGGATTGTCGACACA |
| Cluster-6206.102193 | GCAGCAGTTCCTGGTATGGT | CGCTCGTTTTCGGCTTCTTC |
| Cluster-6206.20086 | TGCTCAGCCAAGAGTAAGCC | CCAAAAGAGGCCGCTGTAGA |
| Cluster-6206.31144 | ACTTTGCCACCGAGGATCAG | ACTGAAAGCAAGCCCCTTGT |
| Cluster-6206.60735 | CGAGCTATTAACGCAGCACA | TGAAATGCCAATGACAAAGC |
| Cluster-6206.73361 | AAGTCACCATTCGAGGGCTG | GTTGAAGTGCGGGACCAAAC |
| Cluster-6206.77187 | ACTGAAAAACTTGCCGTGGC | TACAACTCCACTGCACTGGC |
| Cluster-6206.86111 | TCTGAAACGAGGCTCGAACC | ACCATATAGCAACCTGGCGG |
| Cluster-6206.80232 | GATGCTTGAAACAGTGGCGG | AGTGCTTTAATCCAGCCCCC |
| β-Actin | AGTTTGGTTGATGTGGGAGAC | TGGCTGAACCCGAGATGAT |

*2.6. Data Processing and Analysis*

The data statistics and graphing were performed with graphpad prism, and the data analysis was performed with SAS software (8.0), in which the ANOVA program was used for the variance analysis and Duncan's new multiple range test was used for multiple comparisons. The differential gene heatmap was drawn with TBtools software (11.0.6).

**3. Results and Analysis**

*3.1. Effects of Foliar Mixed Nutrient of Calcium and Magnesium on Pericarp Colouration and the Contents of Sugar and Acid in Pulp*

3.1.1. Change Trend of the Content of Soluble Sugar in Pulp

As shown in Figure 1, the change trends of the contents of the soluble sugar in the pulp were similar to each other between the Ca+Mg treatment and the control in two years. On the 42nd day after flowering, the Ca+Mg treatment exhibited a greater increase than the control, exhibiting an extremely significant difference from the control. After the 56th day after flowering, the Ca+Mg treatment did not change significantly, while the control decreased significantly. After the 63rd day, the Ca+Mg treatment was significantly or extremely significantly higher than the control. This indicated that the Ca+Mg treatment could improve the accumulation of soluble sugar in pulp and make the pulp avoid "sugar withdrawal" after the 56th day after flowering, namely, the issue of "sugar withdrawal" in the pulp was effectively resolved by the Ca+Mg treatment, as it did not even occur.

3.1.2. Chang Trends of the Content of Titratable Acid in Pulp

As seen in Figure 2, the change trends of the contents of titratable acid in the pulp were consistent with each other between the Ca+Mg treatment and the control in two years. During the period from the 42nd to the 49th day after anthesis, the Ca+Mg treatment was extremely significantly lower than the control. Neither the Ca+Mg treatment nor the control changed significantly during the period from the 56th to the 70th day after anthesis. All results indicated that the content of titratable acid in pulp undergoing the Ca+Mg treatment decreased before the harvest, and there was no significant difference between Ca+Mg treatment and the control in the later stage of the fruit development period, so the

flavor quality of 'Feizixiao' litchi pulp was decided mainly by the content of soluble sugars at maturity.

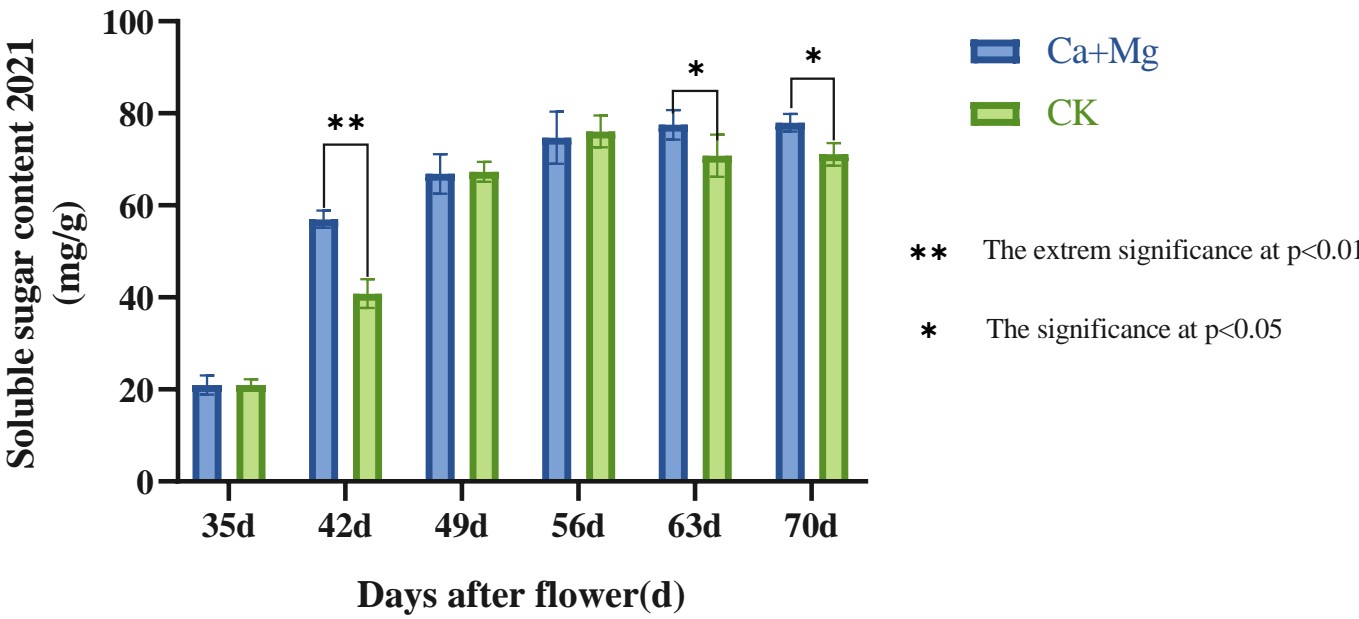

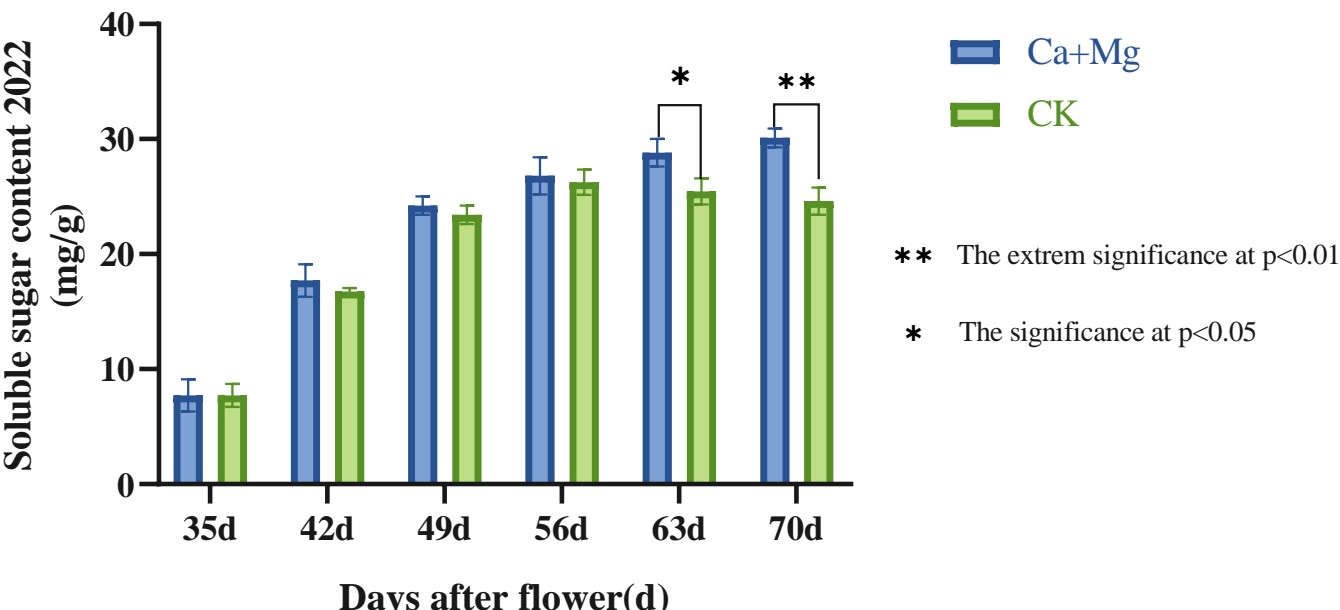

**Figure 1.** Changes in the content of soluble sugar in pulp after the foliar nutrient.

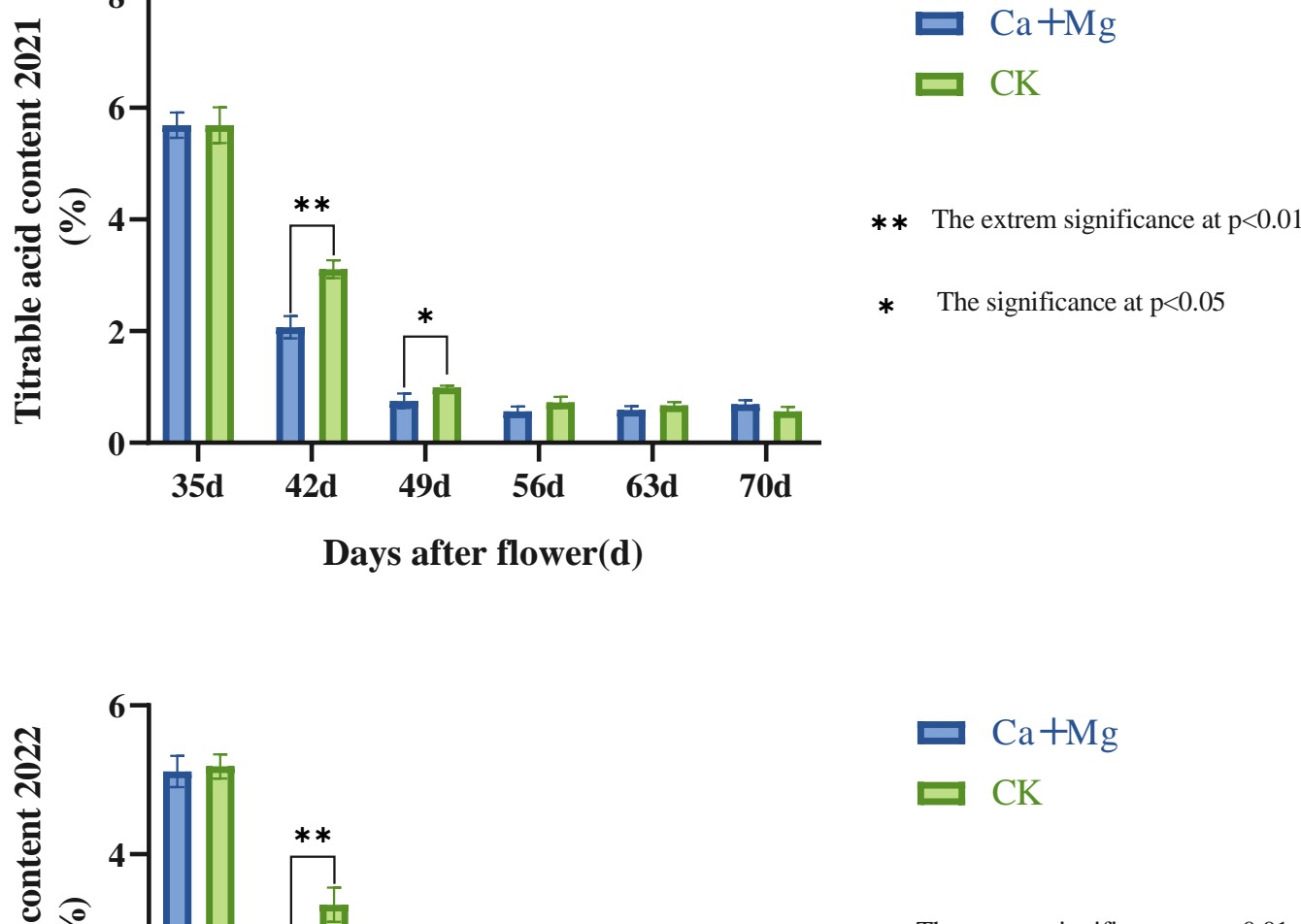

**Figure 2.** Changes in the content of titratable acid in pulp after foliar nutrient.

3.1.3. Effect of Foliar Mixed Nutrient of Calcium and Magnesium on the Coloration of Litchi Pericarp

As shown in Figure 3, the trends of the h value showed a significant difference between the Ca+Mg treatment and the control in two years, and they both decreased after the 42nd day after flowering, namely, the pericarp of both began to color red. On the 49th and 56th day after flowering, the Ca+Mg treatment and the control made no significant difference in the h value. After the 63rd day after flowering, the Ca+Mg treatment was significantly lower than the control, that is, the pericarp of the Ca+Mg treatment was colored better. Therefore, the Ca+Mg treatment improved the pericarp coloration of 'Feizixiao' litchi.

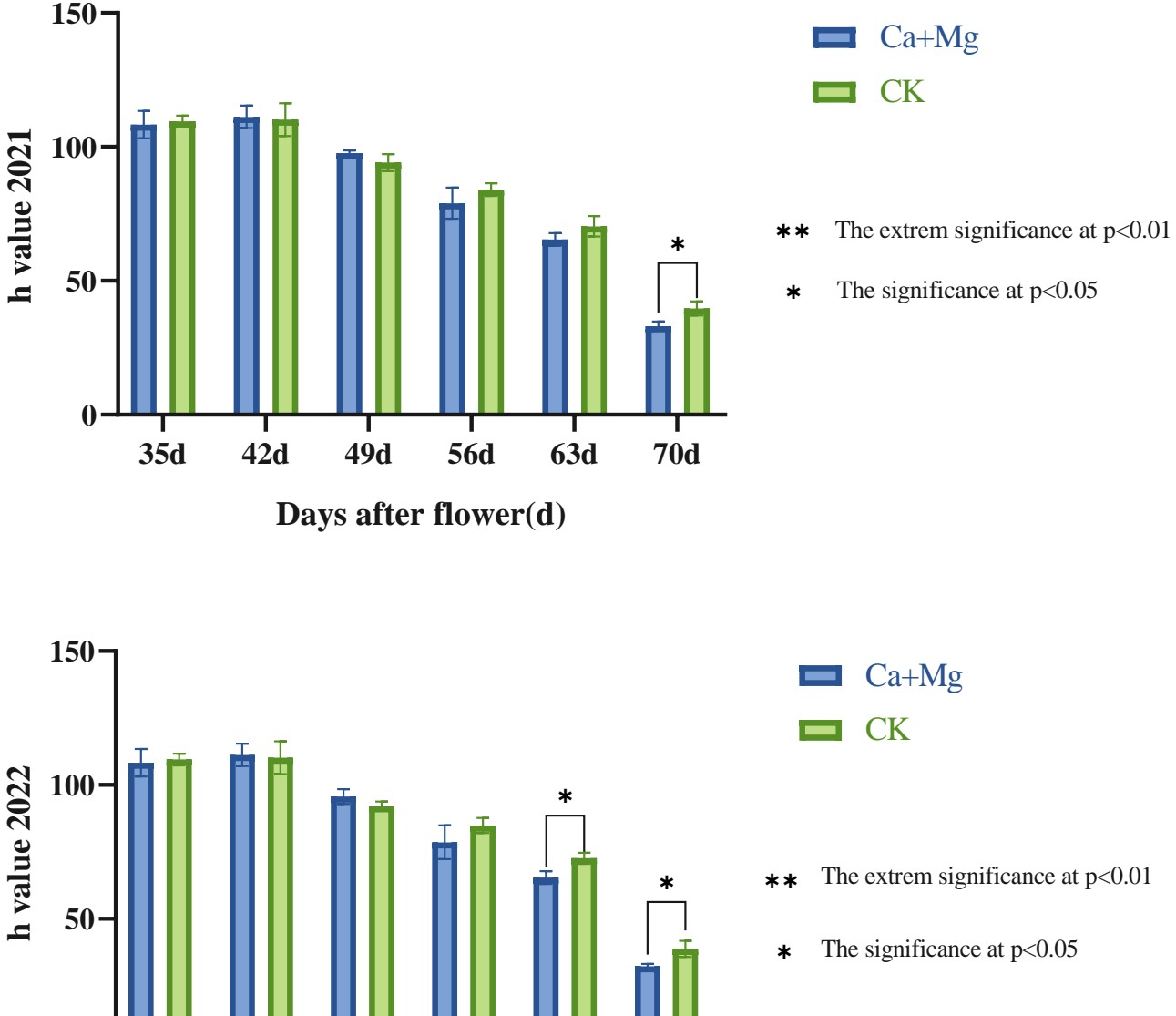

**Figure 3.** Change trend in h value in pulp after foliar nutrient.

*3.2. Effects of Mixed Nutrient of Calcium and Magnesium on the Respiration Rate of the Bypass Pathway*

3.2.1. Trends in the Cytochrome Pathway (CP) Respiration Rates

In Figure 4, there appears a significant difference in the change trends of the CP respiratory rate between the Ca+Mg treatment and control in two years. The Ca+Mg treatment showed a significant increase in the early stage and no significant change in the later stage, while the control showed a trend of first increasing and then decreasing. From the 42nd to 56th days after flowering, the control was significantly or extremely significantly higher than the Ca+Mg treatment. On the 63rd day after flowering, there existed no significant difference between the Ca+Mg treatment and the control; on 70th day after flowering, the Ca+Mg treatment was significantly higher than the control. It can be seen that the Ca+Mg treatment inhibited the respiratory rate of CP in the early stage, and after the 56th day after flowering, the Ca+Mg treatment promoted the CP respiration rate.

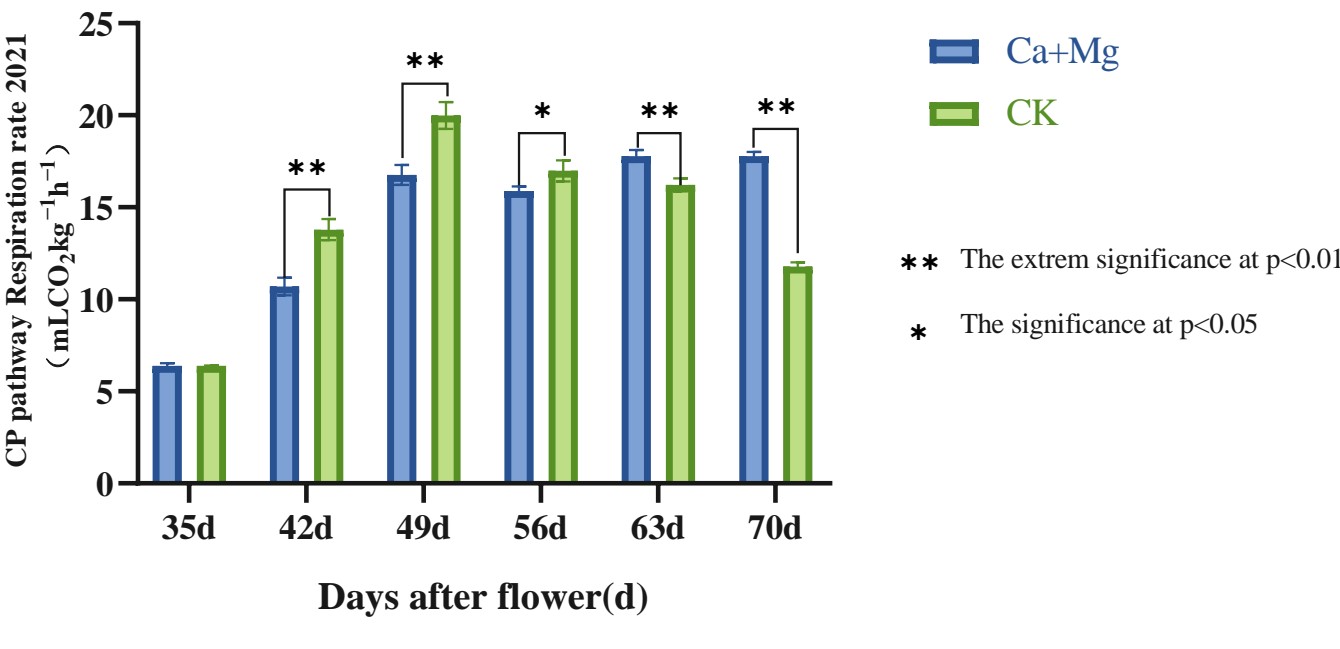

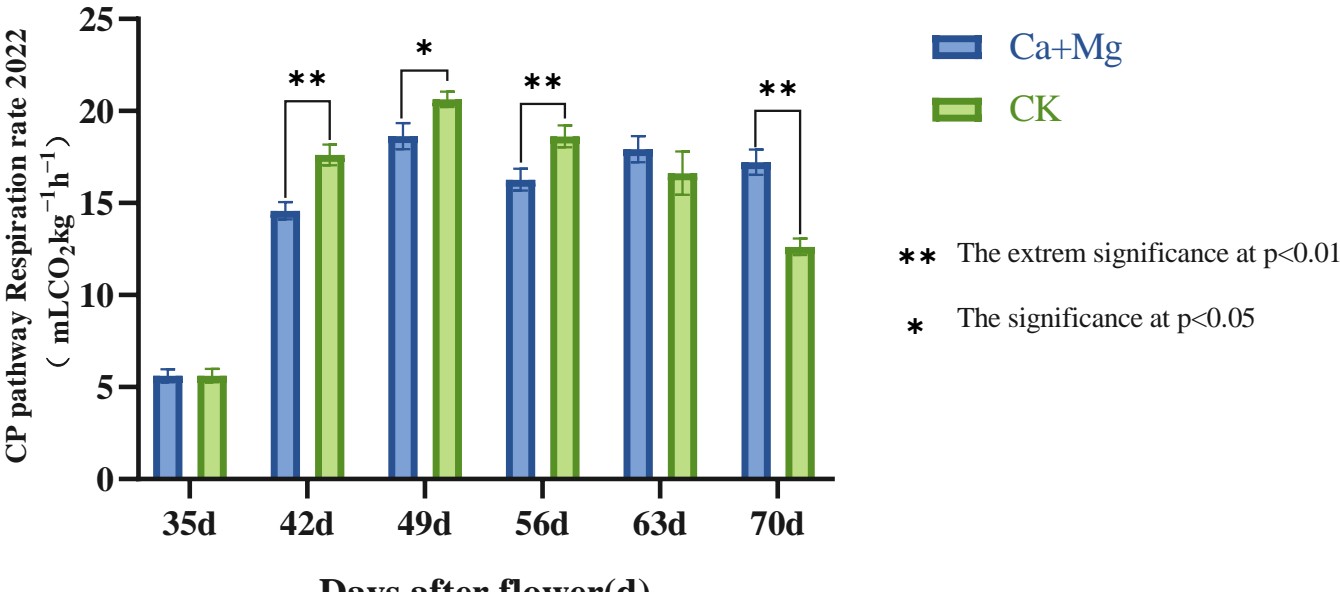

**Figure 4.** Changes in CP pathway respiration rate in pulp after foliar nutrient.

3.2.2. Trends in the Cyanide-Resistant Pathway (AP) Respiratory Rates

As seen in Figure 5, both the Ca+Mg treatment and control showed change trends in the AP respiration rate in the fruit pulp of first increasing and then decreasing. However, the Ca+Mg treatment fluctuated in AP during the period of the near maturity of the fruits. On the 42nd day after flowering, the Ca+Mg treatment was significantly lower than the control. On the 49th day after flowering, the Ca+Mg treatment was significantly lower than the control, or made no significant difference from the control in two years. From the 56th to 70th day after flowering, the Ca+Mg treatment was significantly or extremely significantly lower than the control, or made no significant difference from the control, indicating that the trend of the Ca+Mg treatment was lower than the control in two years. It can be seen that the Ca+Mg treatment inhibited the AP respiration rate.

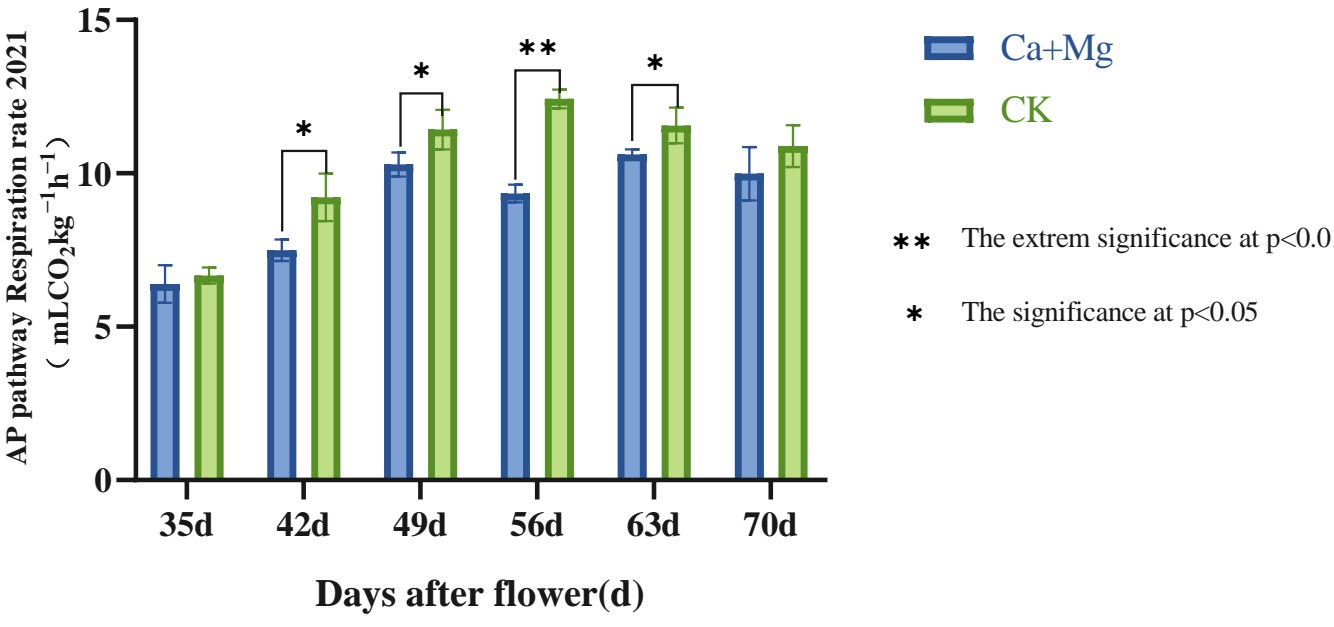

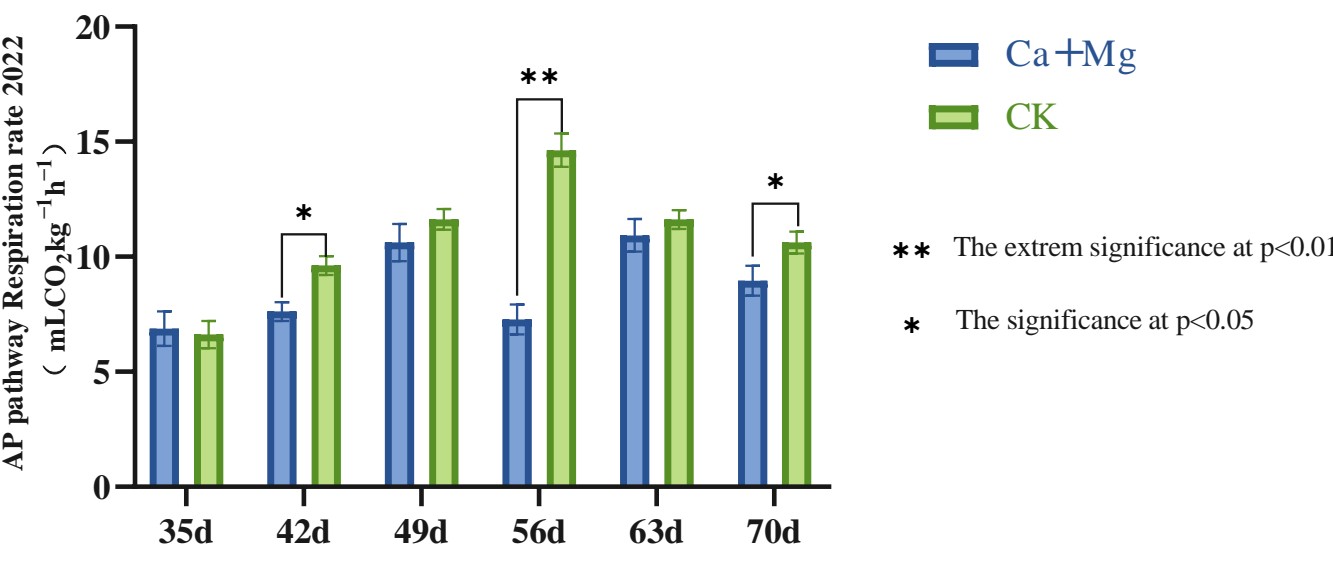

**Figure 5.** Changes in AP pathway respiration rate in pulp after foliar nutrient.

### 3.3. Differential Gene Screening

In order to investigate the effect of bypass respiration on overcoming fruit "sugar withdrawal" with Ca+Mg treatment, a total of 110 genes related to bypass respiratory enzymes were screened from the transcriptome data. Genes with lower expression levels were removed, a heatmap was drawn, as shown in Figure 6, and the expression levels of the Ca+Mg treatment were compared with those of the control during the same period. On the 35th day after flowering, there was no significant difference in the expression levels of related genes between the Ca+Mg treatment and the control. On the 63rd day after flowering, the Ca+Mg treatment up-regulated five genes and down-regulated one gene in the *NADH dehydrogenase* gene family. One gene in the *SDH* gene family was up-regulated and two genes were down-regulated, and three genes in the Cytochrome bc1 complex gene family were up-regulated and one gene was down-regulated. One gene in the *Cyt c* gene

family was up-regulated. Six genes in the *COX* gene family were up-regulated and two genes were down-regulated. All members of the *AOX* gene family were down-regulated. On the 69th day after flowering, the Ca+Mg treatment up-regulated two genes and down-regulated seven genes in the *NADH dehydrogenase* gene family. All members of the SDH gene family were down-regulated. One gene in the *Cytochrome bc₁ complex* gene family was up-regulated and two genes were down-regulated. All members of the *COX* gene family were down-regulated. Two genes in the *AOX* gene family were up-regulated, while one gene was down-regulated.

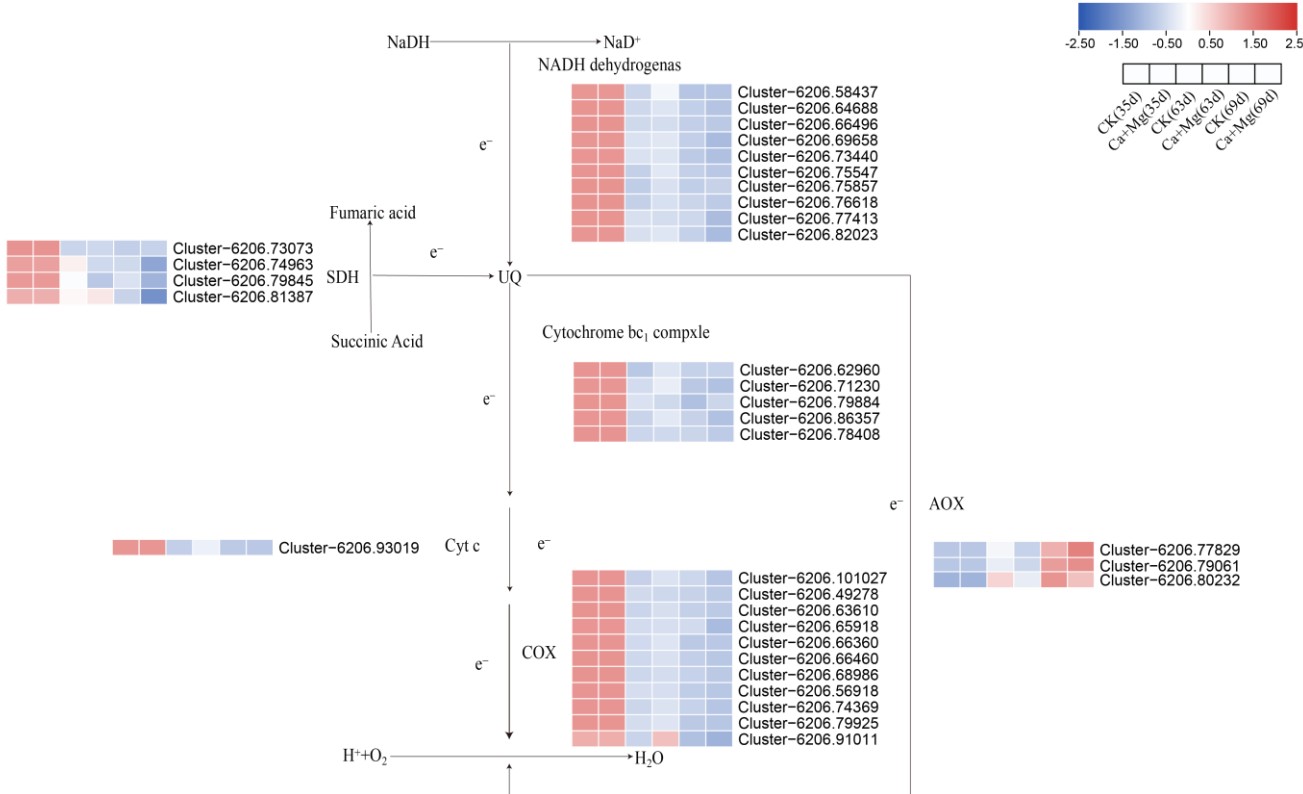

**Figure 6.** Bypass respiratory pathway map and related differential gene expression heatmap.

In summary, after being treated with the mixed foliar nutrient of Ca+Mg, the key enzymes *AOX* and *COX* in the bypass respiratory pathways of 'Feizixiao' litchi pulp jointly participated in inhibiting the "sugar withdrawal" of the pulp on the 63rd day after flowering, while only *AOX* played a role in inhibiting the "sugar withdrawal" of the pulp on the 69th day. The above results show that the CP respiration rate of the Ca+Mg treatment increased in the later stage of fruit growth and development, due to the up-regulation of six and all members of the *COX* gene. At the same time, the AP respiration rate of the Ca+Mg treatment decreased in the later stage of fruit growth and development due to the down-regulation of all and one member of the *AOX* gene family. Therefore, the differential expression of these two key enzyme gene family members in the late stage of fruit growth and development deserves special attention.

*3.4. Real-Time Fluorescence Quantitative PCR Verification*

As can be seen in Figure 7, choosing 10 genes at random, there was a significant one-unit linear correlation between the expression levels detected by real-time PCR and the transcriptome sequencing results ($R^2_{2022} = 0.8275$), thus proving the reliability of the transcriptome sequencing results.

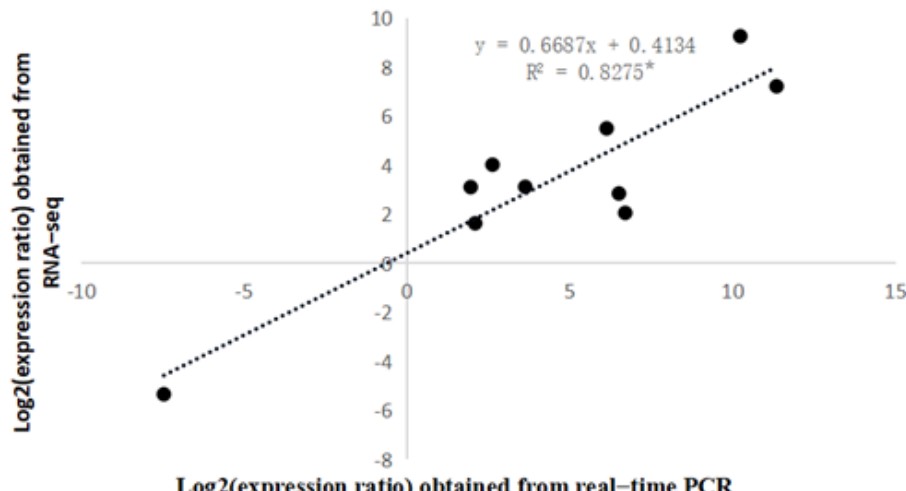

**Figure 7.** One-unit linear correlation analysis between qRT-PCR and transcriptome data. * The significance at $p < 0.05$.

## 4. Discussion

### 4.1. Effect of Different Treatments on 'Feizixiao' Litchi Fruit Quality

Flavor is an important factor for evaluating pulp quality, and the contents of soluble sugar and acid are two important evaluation indicators [38]. The results of this experiment showed that the Ca+Mg treatment improved the edible quality of 'Feizixiao' litchi pulp and overcame the phenomenon of "sugar withdrawal" in its pulp, which is consistent with the results of field experiments conducted by our research group in multiple locations over the years [4,5,39]. Spraying the mixed fertilizer of calcium and magnesium on the leaves during the swelling period of 'Feizixiao' litchi fruit can be used in production to overcome the "sugar withdrawal" in the pulp of 'Feizixiao' litch.

### 4.2. Effects of Different Treatments on the Respiration Rate of the Pulp Electron Transport Chain Pathway

For non-climacteric fruits, entering the senescence stage is always accompanied by a decrease in the CP pathway respiration rate and an increase in the AP pathway. With an increase in AP respiration rate, leading to a decrease in intracellular energy synthesis, the membrane potential is disrupted and fruits enter an aging state [40]. The results of this study indicated that, as non-climacteric fruits [41], Ca+Mg treatment had an inhibitory effect on the CP respiration rate in the early stage and a certain promoting effect on the CP respiration rate during the harvest period, inhibiting the respiratory rate of the AP. Therefore, during the harvesting period, Ca+Mg treatment slowed down the decrease in the respiratory rate of the CP, thereby slowing down the increase in the respiration rate of the AP, delaying the aging process of the fruits.

The Ca+Mg treatment down-regulated *COX1* expression before the 63rd day after flowering, but significantly increased *COX1* expression at the 70th day after flowering. This indicates that the Ca+Mg treatment up-regulated *COX1* expression, thereby increasing *COX* enzyme activity and increasing the CP respiration rate. The Ca+Mg treatment down-regulated the expression of the *AOX* gene, and this down-regulation became more pronounced on the 63rd day after flowering, indicating that the Ca+Mg treatment inhibited *AOX* enzyme activity by inhibiting *AOX* gene expression, thereby reducing the respiratory rate of the AP. In summary, Ca+Mg treatment increases the respiration rate of the CP pathway, ultimately delaying fruit senescence and overcoming the problem of "sugar withdrawal" in 'Feizixiao' litchi.

## 5. Conclusions

During the swelling stage of the 'Feizixiao' litchi, the mixed foliar nutrients of calcium and magnesium can be used in production to overcome the problem of "sugar withdrawal" in the pulp of the 'Feizixiao' litchi.

During the fruit swelling period of the 'Feizixiao' litchi pulp, on the one hand, the mixed foliar nutrients of calcium and magnesium up-regulated the expression of *COX* genes in the pulp and enhanced the activity of *COX* enzymes in the pulp, thereby increasing the respiration rate of the CP. On the other hand, they down-regulated the expression of the *AOX* gene in the fruit pulp, reducing the activity of the *AOX* enzyme and AP respiration rate. Based on these two aspects, the mixed foliar nutrients of calcium and magnesium increased the ATP content of "Feizixiao" litchi pulp during the harvest period, thereby delaying the aging process of the pulp and overcoming the phenomenon of "sugar withdrawal" in the pulp.

At present, only the inhibition of cyanide-resistant respiration and promotion of cytochrome pathways have been determined, and their mechanisms of occurrence are not clear. Further research is needed to investigate the signal transduction pathways triggered by experimental processing and the transcriptional regulation mechanisms of the *COX* and *AOX* genes.

**Author Contributions:** Agreement to be accountable for all aspects of the work in ensuring that questions related to the accuracy or integrity of any part of the work are appropriately investigated and resolved, K.Z.; revising it critically for important intellectual content; final approval of the version to be published, W.M.; Drafting the work; the acquisition, analysis, and interpretation of data for the work, S.S.; the acquisition of data for the work, J.D.; the acquisition, interpretation of data for the work, J.P. All authors have read and agreed to the published version of the manuscript.

**Funding:** National Natural Science Foundation of China (NSFC) (No. 31960570).

**Data Availability Statement:** Data are contained within the article.

**Conflicts of Interest:** The authors declare no conflict of interest.

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
