# Peer review of "The Effects of Mixed Foliar Nutrients of Calcium and Magnesium on the Major Bypass Respiratory Pathways in the Pulp of ‘Feizixiao’ Litchi"

_horticulturae, doi:10.3390/horticulturae10030248_

Round 1

Reviewer 1 Report

Comments and Suggestions for Authors

Dear Authors,

The corrections are described in the attached text.

Best regards,

Comments on the Quality of English Language

Dear Authors,

The corrections are described in the attached text.

Best regards,

Author Response

Dear referee:

Hello! I thank the referee for pointing out shortcomings in the paper. Below are our responses to the reviewers. The reviewers’ comments are in italics,and our responses are numbered.

Q1:I suggest a more detailed description of the tree selection criteria and justification for the specific concentrations of CaCl2 and MgCl2 used in the treatments.

Response:The selected individual plant has moderate growth, uniform tree shape, no mechanical damage, no pests and diseases, and high yield. The results of many years of field experiments in the early stage showed that the optimal concentration of CaCl2 and MgCl2 on the surface of leaves was 0.3%. Higher concentrations of calcium and magnesium would affect the quality of trees and fruits. The experimental design section supplemented this.

Q2: However, a deeper discussion on the selection of genes for qPCR validation and the relationship of the identified genes with specific physiological mechanisms in lychee pulp could enrich the results.

Response: The identified genes are key genes for bypass respiration, which inhibit cyanide resistant respiration and promote cytochrome pathways, making fruits rich in ATP and delaying fruit aging. This is reflected in the post harvest physiology of plant fruits.

Q3: It would be beneficial to discuss more thoroughly how these results compare with previous studies and the potential mechanism by which foliar nutrients affect alternative respiratory pathways.

Response: Previous studies have not investigated the relationship between crop fertilization and bypass respiration, and this information is rare. So I'm sorry, the reviewing teacher. It's difficult to explain the potential mechanism by which leaf fertilizer affects it

Q4: Although the article presents promising data, a section on the study's limitations and possible future research directions could provide a more comprehensive view. For example, investigating the effects of different concentrations of foliar nutrients or their application at other fruit development stages could offer additional insights.

Response: Yes, we added an explanation at the end of the conclusion section. At present, only the inhibition of cyanide resistant respiration and promotion of cytochrome pathways have been determined, and their mechanisms of occurrence are not clear. Further research is needed to investigate the signal transduction pathways triggered by experimental processing and the transcriptional regulation mechanisms of COX and AOX genes.

Q5: The article is well organized, but clarity could be improved in some sections. I recommend a careful review to ensure all parts of the study are presented clearly and concisely, facilitating reader understanding.

Response: The conclusion section has been streamlined to facilitate reader reading.

Q6: Check that all references are up-to-date and that new relevant studies, published after this manuscript's preparation, could be incorporated to strengthen the context and relevance of the study.

Response: We have checked that all references are up-to-date.

Once again we would like to thank the reviewers for their very useful input and we also found your summary most helpful.

Sincerely.

2024/2/29

Reviewer 2 Report

Comments and Suggestions for Authors

In the present manuscript entitled “Effects of Mixed Foliar Nutrients of Calcium and Magnesium 2 on the Major Bypass Respiratory Pathways in the Pulp of 3 'Feizixiao' litchi”, the authors performed an interesting study providing new insights  on how to prevent problems related to "stagnant greenness" of the pericarp, overcome the phenomenon of "sugar withdrawal" in the pulp, and finally to promote CP and inhibit AP, and thus delay the ripening and senescence of the pulp of 'Feizixiao' litchi.

The experimentation was well performed and the methods adopted as the Biolog Eco microplate technology are reliable and effective.

The work is original and can be accepted if the authors respond to the following reviewer's comments:

I would appreciate it if the authors would better explain and emphasize the climacteric, nonclimacteric, or nonrespiratory behavior of lychee, an important aspect for better reader understanding.

The authors address the ripening process of litchi by reducing the stagnant greenness of litchi, the loss of pulp sugars during the ripening phase of the fruit and the subsequent senescence phase.

Through foliar administration of Mg and Ca for each aspect mentioned above, the authors provide an answer:

A)     Mg foliar nutrition can simplify the "stagnant green" problem of the pericarp of lychee 'Feizixiao' after fruiting, through increasing the content of ABA in the pericarp and improving the activity of UFGT, and thus the accumulation of anthocyanins.

B)      Foliar feeding of Ca and Mg mixture inhibits aerobic respiration of the EMP-TCA pathway

through decreasing the activities of GPI, SDH, and NADP-ME and flux of hexoses to EMP and PPP in the pulp of 'Feizixiao' Litchi, thus reducing sugar loss in the pulp.

C)       Based on the principle that higher ATP content is advantageous for delaying fruit senescence, higher COX enzyme activity and CP pathway respiration rate are favorable for ATP synthesis. In addition, an inhibition of AOX enzyme activity through inhibition of AOX gene expression as a result of foliar Ca and Mg administration reduces the AP respiration rate ultimately delaying fruit senescence.

All these assertions are supported by the wealth of experimental data represented in the paper, (particularly on genetic, chemical investigations of sugar content and chemical/physical investigations of CP and AP respiration rates) so it is difficult to refute them.

The only comment I can make is that biologically, in ripening edible fruits, these characteristics occur in the so-called ripening phase involving changes in wall structure; synthesis and accumulation of pigments, synthesis of volatile aromatic compounds, and conversion of starch to soluble sugars.

Therefore, I ask the authors whether ethylene is involved in the regulation of the ripening phase (which I do not believe), that is, the 'increase in respiration, called climacteric peak. I also ask the authors what they mean by non-respiratory climacteric fruit.

I strongly advise the authors to enrich the introduction of the article with these basic concepts to further the reader's understanding.

Finally, I ask the authors, since the climacteric peak is due to a sharp increase in cyanide-resistant respiration, in order to prevent the accumulation of reducing power that could lead to the formation of ROS in the mitochondrion and thus trigger a sequence of oxidative processes that could irreparably damage fruit tissues, is it possible to imagine that foliar treatment with Ca and Mg could generate side effects?

Comments on the Quality of English Language

Moderate English language modification is required

Author Response

Dear Reviewer:

Hello! Thank you for reviewing my paper. Below, I will answer the questions you have raised;

The term "non respiratory climatic fruits" mentioned in papers may have translation errors, and in the vast majority of articles, this type of fruit is referred to as "non climatic fruits"; During the ripening process of fruits, they are divided into "climacteric fruits" and "non climacteric fruits" based on the trend of respiratory rate changes. The climacteric fruits are characterized by a rapid increase in respiratory rate, usually accompanied by an explosion of ethylene; Non respiratory climacteric fruits do not exhibit significant respiratory peaks during their ripening process (Sara Zenoni et al., 2023).

In the study of the ripening process of non climacteric fruits, ABA is considered a key factor in the ripening of non climacteric fruits (Tingting Gu et al., 2019), and no significant burst of ethylene release was detected during the ripening process (Tong Chen et al., 2020). In the study of the ripening process of non climacteric model plants, the changing trends of growth regulatory substances in strawberries and grapes were measured, The trend of ethylene content changes between the two is not consistent, but it is also believed that ethylene participates in the ripening process of this type of fruit through its interaction with ABA (Mar í a Florencia Perotti et al., 2023). In the coloring of lychee peel, ethylene promotes chlorophyll degradation and affects peel coloring (Wang Zhan, 2018). Therefore, ethylene at least participates in the regulation of peel coloring during the ripening process of lychee.

The main purpose of this article's experimental treatment is to regulate fruit ripening and aging development, in order to avoid fruit pulp sugar loss. The question you raised reminds us to pay attention to preventing adverse conditions in the future fertilization technology maturation research. In the Hainan production area, the Concubine Laughter twig is mostly in the dry season from fruiting to maturity, and is most likely to encounter drought. During this period, we need to cooperate with good water management. Other weather disasters are generally not encountered during this period.

Once again we would like to thank the reviewers for their very useful input.

Sincerely

2024/2/29

  • Zenoni S,Savoi S,Busatto N, et al. Molecular regulation of apple and grape ripening: exploring common and distinct transcriptional aspects of representative climacteric and non-climacteric fruits. J Exp Bot. 2023;74 (20):6207-6223.
  • Gu T,Jia S,Huang X, et al. Transcriptome and hormone analyses provide insights into hormonal regulation in strawberry ripening. Planta. 2019;250 (1):145-162.
  • Chen T,Qin G,Tian S. Regulatory network of fruit ripening: current understanding and future challenges. New Phytol. 2020;228 (4):1219-1226.
  • Perotti MF,Posé D,Martín-Pizarro C. Non-climacteric fruit development and ripening regulation: 'the phytohormones show'. J Exp Bot. 2023;74 (20):6237-6253.
  • Wang Zhan. Study on the Mechanism of Leaf Spraying Magnesium Fertilizer to Regulate the Skin Coloring of Feizixiao Litchi. 2018. Hainan University, MA Thesis

Reviewer 3 Report

Comments and Suggestions for Authors

An interesting work, although judging by the available literature, the topic has already been discussed many times. I just have some editorial suggestions:

Line 30: I suggest removing "Feizixiao' Litchi and calcium and magnesium" from the keywords, because these words are in the title. It is worth replacing them with others regarding the content of the work to increase the number of potential readers,

Line 148: single should be capitalized - the beginning of a sentence.

Line 158: there should be a comma after “then”, not a dot.

Line 183 – remove the space before the dot.

Line 332. There should be a dot before "spraying".

Line 362 and 371 it should be "sugar withdrawal”

Line 219, 232, 245, 257, 270, 317 – improve English. The phrase “From figure” is some mental shortcut

Author Response

Dear Reviewer:

Hello!I thank the referee for pointing out shortcomings in the paper. As requested, I have prepared a revised version of our manuscript in the hope of addressing the issues you have raised.

Once again we would like to thank the reviewers for their very useful input. We would love to thank you for allowing us to resubmit a revised copy of the manuscript and we highly appreciate your time and consideration.

Sincerely.

2024/2/29
